

# Bilateral movement asymmetries exist in recreational athletes during a 45° sidestep cut post-anterior cruciate ligament reconstruction

Montana Kaiyala[1],[*], J.J. Hannigan[1],[*], Andrew Traut[2] and Christine Pollard[1]

[1] Program in Physical Therapy, College of Health, Oregon State University—Cascades, Bend, Oregon, United States
[2] Program in Kinesiology, College of Health, Oregon State University, Corvallis, Oregon, United States
[*] These authors contributed equally to this work.

## ABSTRACT

Individuals post-ACL reconstruction (ACLR) are at elevated risk for ACL re-injury. While several studies have examined biomechanical asymmetries post-ACLR during landing, less is known about asymmetries during a sidestep cut. Therefore, the purpose of this study was to compare sagittal and frontal plane biomechanics at the hip and knee during a 45° sidestep cut in post-ACLR participants and healthy controls. Nineteen athletes post-ACLR and nineteen healthy controls performed a bilateral 45° sidestep cut while three-dimensional kinematics and kinetics were measured. Sagittal and frontal plane kinematics and kinetics were examined at the hip and knee during stance phase. A linear mixed model compared biomechanical differences between the limbs of ACLR and healthy control participants ($\alpha = 0.05$). In the post-ACLR group, peak hip extension, peak knee flexion, sagittal hip and knee excursion, and the peak knee extensor moment were significantly lower in the ACLR surgical limb compared to the non-surgical limb ($p < 0.05$). The peak knee flexion angle and peak knee extensor moment were also lower in the ACLR surgical limb compared to the matched control limb ($p < 0.05$). In summary, post-ACLR participants exhibited altered sagittal plane movement in their surgical limb that was not demonstrated in the non-surgical limb or in control participants, which may suggest avoidance, or reduced utilization of the ACLR limb.

# INTRODUCTION

Anterior cruciate ligament (ACL) rupture is a potentially devastating injury that affects an estimated 250,000+ individuals each year, with those primarily affected being young, physically active, or female (*Griffin et al., 2006*; *Paterno et al., 2012*). Individuals who sustain an ACL injury may experience physical and psychological barriers as they return to full physical activity and sport and have an elevated risk of developing knee osteoarthritis (OA) (*Vutescu et al., 2021*; *Everhart, Yalcin & Spindler, 2021*; *Webster & Hewett, 2022*).

Corresponding author
J.J. Hannigan,
hannigaj@oregonstate.edu

The gold standard treatment for an ACL rupture in active individuals is surgical ACL reconstruction (ACLR) followed by a recommended 9 to 12 months of physical rehabilitation. At the conclusion of rehabilitation, patients are generally cleared to return to sport (RTS) and full physical activity if they are able pass a battery of RTS tests. This battery often involves a series of dynamic landing tasks, with success often defined as 90–100% movement symmetry between the involved (surgical) and uninvolved (non-surgical) limbs (*Gokeler, Dingenen & Hewett, 2022*). These tests are designed to protect athletes from premature RTS and secondary ACL injury (*van Melick et al., 2016*). However, re-injury rates post-ACLR and RTS may be as high as 33% (*Wright et al., 2011*; *Paterno et al., 2012*, *2014*; *Kaeding et al., 2015*; *Wiggins et al., 2016*; *Lai et al., 2018*; *Nawasreh et al., 2018*; *Webster, Feller & Klemm, 2021*). In addition, athletes post-ACLR are up to 15 times more likely to experience a future ACL rupture compared to individuals with no history of ACL injury (*Paterno et al., 2012*). This high rate of re-injury suggests that current rehabilitation protocols may still need improvement, with a specific focus on preventing future ACL injuries.

ACL injuries can occur *via* contact or non-contact mechanisms, with non-contact mechanisms constituting approximately 70% of all ACL injuries in females (*Kobayashi et al., 2010*). The majority of non-contact ACL injuries occur when an athlete is decelerating, changing directions, or landing (*Boden et al., 2000*). An athlete's risk of ACL injury or re-injury may be related to aberrant biomechanical movement patterns during these types of tasks (*Hewett et al., 2005*; *Paterno et al., 2010*).

Historically, the majority of research examining post-ACLR biomechanics has focused on landing tasks, with fewer studies incorporating change of direction (COD) tasks. During the years following primary ACLR and RTS, current research suggests athletes may exhibit altered biomechanical patterns during dynamic jump-landing tasks (*Paterno et al., 2007*, *2010*; *Trigsted, Post & Bell, 2017*). These altered patterns include reduced knee flexion at initial contact and reduced hip and knee extensor moments in the ACLR limb compared to both the non-surgical limb and to healthy control limbs (*Trigsted, Post & Bell, 2017*; *Lepley & Kuenze, 2018*). These findings have been associated with an elevated risk of ACL re-injury (*Paterno et al., 2010*).

Fewer studies have investigated tasks that involves both deceleration and change-of-direction post-ACLR, such as a sidestep cut, despite research suggesting that the majority of non-contact ACL injuries occur during such a task (*Boden et al., 2000*), and that this task may be the most sensitive to reveal post-ACLR asymmetries (*Kotsifaki et al., 2022*). A recent investigation by *King et al. (2019)* found that during a planned 90° COD maneuver, individuals post-ACLR demonstrated greater asymmetry than healthy controls in their hip abductor moment, with no differences noted in the sagittal plane. Using the same task, *Clark et al. (2019)* reported greater knee valgus in the ACLR limb in 8 of 10 post-ACLR athletes. *Stearns & Pollard (2013)* discovered that, during a 45° sidestep cut, post-ACLR individuals demonstrated a greater peak knee valgus angle and peak knee varus moment at the knee compared to healthy controls but did not examine bilateral symmetry or hip biomechanics.

Compared with a 90° sidestep cut, a 45° sidestep cut generally requires significantly different lower extremity joint kinematic and kinetics (*Havens & Sigward, 2015*; *Schreurs, Benjaminse & Lemmink, 2017*) and may better replicate common sports movements. Comparing cutting biomechanics between the surgical limb in ACLR patients and both their non-surgical limb as well as a healthy control limb provides the most comprehensive data to illustrate whether biomechanical deficits still exist after rehabilitation and RTS. To our knowledge, no single study to date has investigated both between-limb differences and ACLR participant compared to healthy matched control differences during a cutting task. Therefore, the purpose of our study was to compare frontal and sagittal plane hip and knee joint kinetics and kinematics during a 45° sidestep cut in individuals post-ACLR and in healthy controls. It was hypothesized that (a) post-ACLR individuals would cut with a greater peak knee valgus angle and peak knee varus moment in their surgical limb compared to their non-surgical limb and the matched control limb, and (b) control participants would not demonstrate any frontal or sagittal plane between-limb differences.

## MATERIALS AND METHODS

### Participants

IRB approval (Oregon State University #7995) and written informed consent from all participants were obtained prior to participation in the study. Participants were recruited *via* fliers, social media, and word-of-mouth. For all participants, eligibility criteria included being between 18–55 years old, participating in at least 30 min of physical activity including cutting or jumping three times per week, having no new injuries within the last 6 months to the lower extremities or back, and not suffering from a neurological or vascular disorder that affects participation in sport. For the ACLR group, participants needed to have their ACL surgically reconstructed at least 12 months prior to participation, be cleared for unrestricted physical activity by their physical therapist or orthopedic surgeon, and have no previous orthopedic surgeries to their lower extremities or back.

Based on previous unpublished data collected in our laboratory (data is available here: https://www.doi.org/10.17605/OSF.IO/SZKW2), 17 participants per group were needed to adequately power the study (power = 0.8), calculated using G*Power (Düsseldorf, Germany). To account for additional variability, 19 individuals post-ACLR and 19 control participants who had not undergone an ACLR were included in the study. Participant demographics can be seen in Table 1. ACLR participants had a mix of graft types (two patellar tendon, eight hamstring, five cadaver, and four hamstring/cadaver). The average time since ACLR was 54.9 ± 45.2 months. Two of the 19 ACLR participants had undergone two ipsilateral ACLR surgeries. Control participants had no history of ACL injury and were matched with a specific ACLR participant based on demographic measures of sex, height, mass, age, and self-reported activity level. Once each ACLR participant was matched to a control participant, the side of the surgical limb in the ACLR participant (right or left) was chosen as the matched limb for the between-subjects comparison (*i.e.*, if the ACLR participant's surgical side was the right limb, the right limb of their matched control participant was chosen as the comparison limb).
**Table 1 Participant demographics.**

| Variable | ACLR group | Control group | p-value | t | df |
|---|---|---|---|---|---|
| Sex | 10F, 9M | 10F, 9M | | | |
| Age (years) | 31.9 ± 9.6 | 27.5 ± 7.7 | 0.13 | 1.54 | 36 |
| Height (m) | 1.7 ± 0.1 | 1.8 ± 0.1 | 0.14 | 1.50 | 36 |
| Weight (kg) | 72.8 ± 13.7 | 71.0 ± 11.8 | 0.68 | 0.42 | 36 |
| Activity level (min/week) | 381.1 ± 276.2 | 374 ± 172.1 | 0.93 | 0.92 | 36 |

## Instrumentation

Kinematic data were collected using a Vicon eight-camera, three-dimensional motion capture system (Oxford Metrics LTD., Oxford, England) at a sampling frequency of 250 Hz. The cameras were interfaced to a computer and positioned to capture the area around two floor-embedded force plates (Advanced Mechanical Technologies, Inc., Newton, MA, USA). The force plates (1,000 Hz) were interfaced to the same computer that was used for kinematic data collection *via* an analog to digital converter and synchronized with the motion capture data.

## Procedures

Participants were given laboratory shorts and shoes to wear for the data collection. The shoes were a traditional New Balance running shoe (NB 880v2). Reflective markers (14 mm spheres) were placed bilaterally over the following anatomical landmarks: the 1st and 5th metatarsal heads, distal interphalangeal joint of the 2nd toe, medial and lateral malleoli, medial and lateral femoral epicondyles, greater trochanters, iliac crests, anterior superior iliac spines (ASIS), and the joint space between the fifth lumbar and the first sacral spinous processes. Rigid clusters of four reflective tracking markers were attached bilaterally to the participant's thigh and leg. In addition, rigid clusters of three reflective tracking markers were placed bilaterally on the heel counter of the shoe. This marker set has been used in many previous studies (*Hannigan & Pollard, 2019*, *2020*). Marker placement consistency was ensured by having a highly trained and experienced researcher place the markers on each participant. Following marker placement, a static calibration trial was collected before all markers were removed except for the five markers on the pelvis and the six tracking clusters.

An aerial diagram of the cutting pathway and capture space is included in Fig. 1. From a standing position, participants were instructed to run seven meters down a runway at a speed of 3.35 m/s before contacting the force platform and performing a 45° sidestep cut in the direction opposite of their planted limb. Participants were instructed before each trial which limb to perform the cut and were allowed to target the force plate or chop their steps before cutting to best simulate a sports environment. Cutting pathways were established in both directions (right and left) by cones placed at 35° and 55° from the original direction of progress. This aided the consistency of the desired 45° cutting angle. Speed was determined by timing gates placed at the start of the runway and at the end of the cutting pathway

(Fig. 1). Participants were allowed 3–5 practice trials in each direction to become familiar with the procedures, instrumentation, and speed. Participants then completed four successful trials on each limb, with success defined by the following criteria: (a) their foot remained within the borders of the force platform, (b) they remained within the cutting pathway designated by the cones, and (c) they maintained the required approach speed of 3.35 m/s ±5%.

## Data analysis

Coordinate data were digitized in Vicon Nexus software (Vicon Motion Systems Ltd., Oxford, UK). Foot strike was defined as the first frame the vertical ground reaction force was greater than 20 Newtons, while toe-off was defined as the first frame the vertical ground reaction force was less than 20 Newtons. Cutting velocity at the initial foot strike of the cutting maneuver was calculated for each participant using the methods of Vanrenterghem et al. (2010, 2012). Kinematic data were filtered using a fourth-order zero-lag Butterworth 12-Hz low-pass filter. Visual3D software (C-Motion, Inc., Germantown, MD, USA) was used to quantify three-dimensional joint kinematics using a joint coordinate system approach. Kinematics, ground reaction forces and anthropometrics were used to calculate joint moments using inverse dynamics equations in Visual3D. Kinetic data were normalized to body mass. The joint moments referred to in this investigation are internal joint moments.

Peak kinematic and kinetic variables were measured during stance phase which was defined as the time from foot strike to toe-off of the plant limb. Kinematic variables of interest included peak hip extension, peak knee flexion, hip and knee sagittal plane excursion, peak hip adduction, and peak knee valgus. Kinetic variables of interest included peak hip and knee extensor moments, peak hip abductor moment and peak knee varus moment.

## Statistical analysis

A linear mixed model was used to compare data between limbs and between participant groups. In this model, subject ID was inputted as a random effect, while limb (ACLR: surgical, non-surgical, control: matched, unmatched) nested within group (ACLR or control) was inputted as a fixed effect. This statistical approach allows both a between-limb comparison of the same participant and a between-groups comparison between ACLR and control participants without violating the assumption of independence of observations in a traditional mixed effects ANOVA. The reported $p$-values are for the main effect of limb (nested within group) from the linear mixed model using an omnibus alpha-level of 0.05. Follow-up Bonferroni-adjusted pairwise tests were performed for the following comparisons: (a) ACLR surgical limb compared to ACLR non-surgical limb, (b) ACLR surgical limb compared to matched control limb, (c) control matched limb compared to control unmatched limb, and d) ACLR non-surgical limb compared to control unmatched limb. All statistical testing was performed using SPSS version 25 (SPSS Inc., Chicago, IL), except for standardized mean differences (Cohen's $d$), which were calculated using Excel (Microsoft, Redmond, WA, USA).

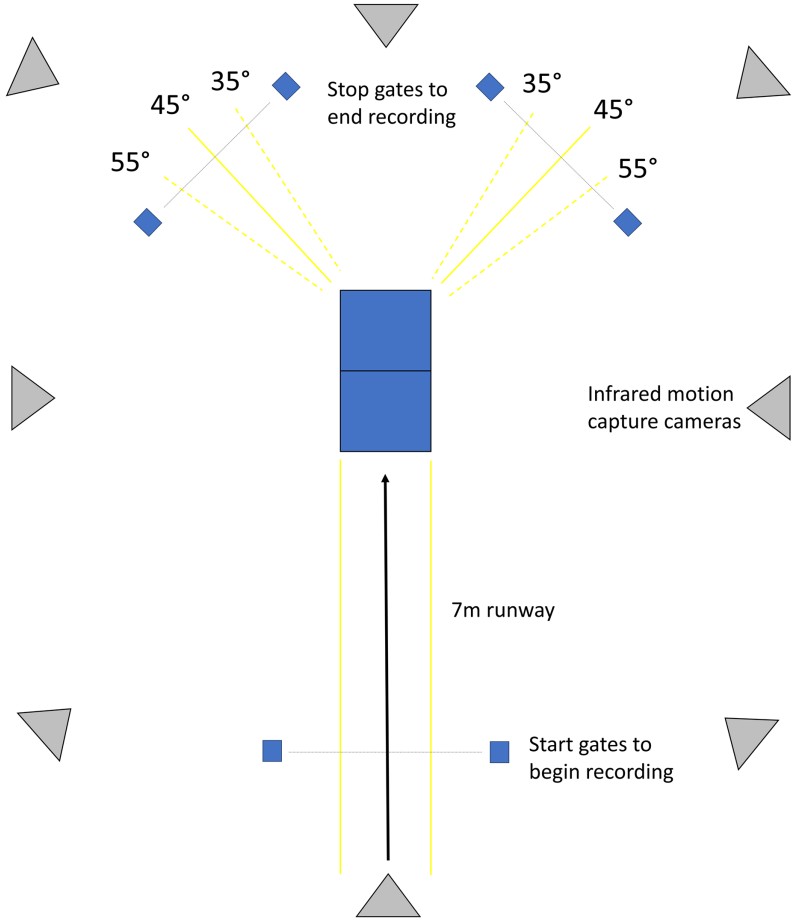

**Figure 1** **An aerial diagram of the cutting pathway and capture space.** Participants ran down the 7-m runway at a speed of 5.5 m/s ± 5%, planted their foot on one of two force platforms, and performed a sidestep cut of approximately 45° in the direction to the opposite of the planted limb. Cones placed at 35° and 55° established a cutting pathway.               

## RESULTS

Cutting velocity at initial foot strike of the cutting maneuver was not different between all limbs, $p = 0.339$ (Table 2).

At the hip, peak hip extension was significantly lower in the ACLR limb relative to the non-surgical limb (ACLR surgical: 11.53 ± 9.26°, ACLR non-surgical: 16.09 ± 11.83°, $p = 0.017$, $F_{3,39} = 3.81$). Hip extension excursion was also lower in the ACLR surgical limb relative to the non-surgical limb (ACLR surgical: 52.84 ± 11.34°, ACLR non-surgical: 58.65 ± 9.16°, $p = 0.021$, $F_{3,44} = 3.58$). No other significant differences were observed between limbs or groups, including for peak hip adduction ($p = 0.630$), the peak hip extension moment ($p = 0.839$), or the peak hip abduction moment ($p = 0.510$) (Table 2). Standardized mean differences for each limb comparison, equivalent to Cohen's $d$, can be seen in Table 3.

At the knee, peak knee flexion was significantly lower in the ACLR surgical limb compared to both the non-surgical limb and the matched limb of the control group (ACLR

**Table 2 Biomechanical comparison between groups and limbs.**

| Variable | ACLR group surgical limb | ACLR group non-surgical limb | Control group matched limb | Control group unmatched limb | *p*-value | F | df |
|---|---|---|---|---|---|---|---|
| Cutting velocity | 3.14 ± 0.32 | 3.22 ± 0.33 | 3.02 ± 0.40 | 3.08 ± 0.36 | 0.339 | 1.16 | 3,40 |
| Hip | | | | | | | |
| *Peak extension* (°) | 11.53 ± 9.26 | 16.09 ± 11.83 | 16.55 ± 8.56 | 15.24 ± 9.14 | 0.017* | 3.81 | 3,39 |
| *Extension excursion* (°) | 52.84 ± 11.34 | 58.65 ± 9.16 | 56.69 ± 8.08 | 54.59 ± 7.99 | 0.021* | 3.58 | 3,44 |
| *Peak adduction* (°) | −0.37 ± 7.28 | 0.26 ± 6.69 | 1.72 ± 4.41 | 2.26 ± 6.00 | 0.630 | 0.58 | 3,49 |
| *Peak extensor moment (Nm/kg)* | 2.49 ± 1.18 | 2.41 ± 1.09 | 2.53 ± 0.97 | 2.60 ± 0.92 | 0.839 | 0.28 | 3,38 |
| *Peak abduction moment (Nm/kg)* | 1.67 ± 0.51 | 1.64 ± 0.44 | 1.64 ± 0.38 | 1.50 ± 0.40 | 0.510 | 0.78 | 3,50 |
| Knee | | | | | | | |
| *Peak flexion* (°) | 44.16 ± 10.20 | 48.20 ± 6.81 | 50.26 ± 7.74 | 51.50 ± 7.82 | 0.013*,† | 4.09 | 3,41 |
| *Flexion excursion* (°) | 26.88 ± 6.55 | 32.12 ± 5.92 | 31.04 ± 7.74 | 31.17 ± 7.35 | 0.002* | 5.78 | 3,42 |
| *Peak valgus* (°) | 8.41 ± 5.52 | 8.53 ± 5.41 | 9.03 ± 4.43 | 8.24 ± 4.75 | 0.930 | 0.15 | 3,49 |
| *Peak extensor moment (Nm/kg)* | 2.61 ± 0.70 | 3.14 ± 0.43 | 3.01 ± 0.45 | 3.27 ± 0.50 | <0.001*,†,# | 10.52 | 3,45 |
| *Peak varus moment (Nm/kg)* | 0.33 ± 0.22 | 0.36 ± 0.21 | 0.42 ± 0.26 | 0.50 ± 0.23 | 0.144 | 1.88 | 3,50 |

Notes:

The F column lists the F-statistic from the linear mixed model. The df column lists the degrees of freedom from the linear mixed model (numerator df,denominator df).

\* A significant difference between the surgical and non-surgical limb in the ACLR group.

† A significant difference between the surgical limb of the ACLR group and the matched limb of the control group.

# A significant difference between the matched and unmatched limbs in the control group.

surgical: 44.16 ± 10.20°, ACLR non-surgical: 48.20 ± 6.81°, matched control: 50.26 ± 7.74°, $p = 0.013$, $F_{3,41} = 4.09$). Knee flexion excursion was significantly lower in the ACLR surgical limb compared to the non-surgical limb (ACLR surgical: 26.88 ± 6.55°, ACLR non-surgical: 32.12 ± 5.92°; $p = 0.002$, $F_{3,42} = 5.78$). Finally, the peak knee extensor moment was significantly lower in the ACLR surgical limb compared to both the ACLR non-surgical limb and matched control limb, and also significantly lower in the control matched limb compared to the unmatched limb (ACLR surgical: 2.61 ± 0.70 Nm/kg, ACLR non-surgical: 3.14 ± 0.43, matched control: 3.01 ± 0.45, unmatched control: 3.27 ± 0.50 Nm/kg; $p = 0.001$, $F_{3,45} = 10.52$). No other significant differences were observed between limbs or groups, including for peak knee valgus ($p = 0.930$), or for the peak knee varus moment ($p = 0.144$). Standardized mean differences for each limb comparison, equivalent to Cohen's *d*, can be seen in Table 3.

## DISCUSSION

Our findings indicate that on average, individuals post-ACLR demonstrated sagittal plane kinematic and kinetic differences at the hip and knee in their surgical limb compared to their non-surgical limb, and in some cases compared to the matched control limb, during the stance phase of a 45° sidestep cut. Specifically, peak hip extension, hip extension excursion, peak knee flexion, knee flexion excursion, and the peak knee extensor moment were significantly lower in the ACLR surgical limb relative to the non-surgical limb, while

**Table 3 Standardized mean differences, equivalent to Cohen's *d*, for limb comparisons of interest.**

| Variable | ACLR surgical limb-ACLR non-surgical limb | ACLR surgical limb-control matched limb | ACLR non-surgical limb-control unmatched limb | Control matched limb–control unmatched limb |
|---|---|---|---|---|
| **Hip** | | | | |
| Peak extension (°) | **0.59** | 0.56 | 0.08 | 0.30 |
| Extension excursion (°) | **0.56** | 0.39 | 0.48 | 0.41 |
| Peak adduction (°) | 0.08 | 0.35 | 0.32 | 0.10 |
| Peak extensor moment (Nm/kg) | 0.16 | 0.04 | 0.19 | 0.12 |
| Peak abduction moment (Nm/kg) | 0.07 | 0.08 | 0.33 | 0.04 |
| **Knee** | | | | |
| Peak flexion (°) | **0.53** | **0.67** | 0.45 | 0.27 |
| Flexion excursion (°) | **0.93** | 0.58 | 0.14 | 0.02 |
| Peak valgus (°) | 0.02 | 0.13 | 0.06 | 0.23 |
| Peak extensor moment (Nm/kg) | **0.91** | **0.69** | 0.27 | **0.73** |
| Peak varus moment (Nm/kg) | 0.11 | 0.39 | 0.64 | 0.33 |

**Note:**
Statistically significant comparisons, as outlined in Table 2, are bolded.

peak knee flexion and the peak knee extension moment were also significantly lower in the ACLR surgical limb compared to the matched control limb. The standardized mean differences for these significant comparisons, equivalent to Cohen's *d*, were between 0.53 and 0.93, indicating moderate to large effects that may have clinical relevance. In contrast, we did not find any differences between limbs for any frontal plane variables. The observed sagittal plane differences, and lack of differences in the frontal plane, did not support our first hypothesis.

Within the control group, outside of the peak knee extensor moment, we did not observe significant differences between matched and unmatched limbs, suggesting that the between limb differences observed in the ACLR group are specific to this population. While we did observe differences in the peak knee extensor moment between limbs in the control group, the peak knee extensor moment in the ACLR surgical limb was significantly lower than the matched control limb, and the combined average of both control limbs (3.14 Nm/kg) was the same as the non-surgical limb in the ACLR group (3.14 Nm/kg). Together, this suggests that the difference observed between limbs for the peak knee extension moment in the control group may be due to random sampling variability and not indicate a difference in stiffening strategy between limbs. Therefore, our second hypothesis is largely supported and agrees with previous research in healthy individuals (*Pollard et al., 2020*).

Overall, our findings suggest that many post-ACLR participants used a stiffening strategy in their ACLR limb in which less range of motion was utilized. This may indicate

avoidance due to muscular weakness, reduced proprioception, altered neuromuscular control, or kinesiophobia (*Nagai et al., 2013*; *Schmitt et al., 2015*; *Trigsted, Post & Bell, 2017*; *Paterno et al., 2018*; *Johnston, McClelland & Webster, 2018*). Although a unilateral stiffening movement strategy theoretically conserves energy for the knee extensor muscles by limiting peak knee flexion during stance phase, it may also expose the knee joint to higher forces and increased ACL strain, which has been demonstrated to increase as the knee flexion angle approaches 30° (*Li et al., 2004*; *Yang et al., 2023*). These factors have been associated with an elevated risk of ACL re-injury (*Paterno et al., 2010*; *Trigsted, Post & Bell, 2017*). Therefore, these findings may be one factor that helps explain why individuals with a history of ACL injury are more likely than individuals without a previous ACL injury to experience a future ACL injury (*Paterno et al., 2012*).

The sagittal plane differences that we observed share some similarities with previous findings during single-leg tasks. During a 90° unplanned cutting task, previous work by King et al. demonstrated that ACLR participants displayed significantly more asymmetry in their knee flexion angle than controls (*King et al., 2019*), with the ACLR limb displaying less knee flexion for the majority of stance phase (*King et al., 2018*). In addition, the significant reduction in peak knee flexion and the peak knee extensor moment in the ACLR limb relative to the non-surgical limb have also been observed during single leg landings (*Oberländer et al., 2013*; *Lepley & Kuenze, 2018*; *Johnston, McClelland & Webster, 2018*). Reduced quadriceps strength in the surgical limb has been suggested as a potential mechanism for these findings, (*Oberländer et al., 2013*; *Schmitt et al., 2015*; *Benjaminse et al., 2019*). Reduced quadriceps strength has been strongly associated with a smaller peak knee extension moment during single-leg hopping (*Oberländer et al., 2013*). If this relationship applies to cutting, the smaller peak knee extension moment we observed may be the result of weaker knee extensors with less capacity to eccentrically control knee flexion, resulting in a smaller peak knee flexion angle. Despite a smaller peak knee extension moment, this knee extended posture may increase quadriceps-induced ACL loads (*Li et al., 2004*; *Yang et al., 2023*) and contribute to ACL re-injury (*Johnston, McClelland & Webster, 2018*). We did not measure quadriceps strength in this study, but based on our current findings, these variables should be investigated in future research.

Our finding that participants used a protective strategy during a sidestep cut is also consistent with similar studies examining double-leg tasks. During a two-legged jump landing, participants post-ACLR have shown lower knee extensor moments and lower vertical ground reaction forces in their ACLR limb relative to their non-surgical limb and healthy controls, indicating that these participants attempted to unload their ACLR limb by shifting demand to the non-surgical limb, possibly for the purpose of resisting quadriceps fatigue or preventing ipsilateral re-injury (*Lepley & Kuenze, 2018*). The present study suggests that during a sidestep cut, because the motion is inherently single-legged and the demand cannot be shifted to the contralateral limb, athletes may instead employ an alternative protective strategy in which they limit their range of motion and knee extensor demand during stance phase.

Surprisingly, we found no asymmetries in frontal plane hip and knee mechanics in the post-ACLR group, nor any differences in frontal plane mechanics between groups, which

contrasts with several previous studies. *Stearns & Pollard (2013)* found that soccer athletes post-ACLR exhibited greater knee frontal plane angles and moments compared to healthy controls during a 45° sidestep cut. In addition, *King et al. (2019)* reported that during a planned 90° COD maneuver, individuals post-ACLR demonstrated greater asymmetry than healthy controls in their hip abductor moment after initial contact. Using the same task as *King et al. (2019)* and *Clark et al. (2019)* discovered that 8 of 10 post-ACLR knees demonstrated greater than 5° of valgus, while only 6 of 10 uninvolved knees reached that threshold. These differences may be due to the difference in task and sample populations, as *King et al. (2019)* and *Clark et al. (2019)* used a 90° cut, and King tested athletes just 9 months post-ACLR. In addition, *Stearns & Pollard (2013)* tested female soccer players post-ACLR, whereas we tested both male and female recreational athletes, and previous research suggests that female athletes are more prone to cut with significant valgus compared to males (*Ford et al., 2005*).

Finally, no significant differences were observed between the non-surgical limb of the ACLR participants and unmatched limb of control participants. Also, as previously discussed, the only difference observed between limbs in the control participants was for the peak knee extension moment, which may be due to sampling variability and not indicative of a physiologic difference. The lack of differences observed for these comparisons provides further support for the validity of our findings relative to the ACLR surgical limb, suggesting that the deficits observed in the surgical limb were physiologic and not happenstance.

A limitation of our study was that our participant pool was relatively heterogenous and included a range of participant age, sex, current activity level, graft type and length of time post-RTS, which may affect movement biomechanics (*Sharafoddin-Shirazi et al., 2020*; *Miles et al., 2022*). In addition, each participant wore standard running shoes instead of more sport-specific footwear such as cleats. The present study would be strengthened if we included a measure of self-reported kinesiophobia, quadriceps strength and proprioception to investigate whether these measures were associated with the identified differences between limbs. As such, further research is needed to determine the factors which contribute to our observed limb differences during a 45° sidestep cut to further inform RTS protocol after ACLR. Future research should also consider adding an unanticipated sidestep cutting condition, as research has demonstrated that this condition results in altered mechanics at the knee compared to an anticipated sidestep cut and may better reflect game-like conditions (*Almonroeder, Garcia & Kurt, 2015*).

## CONCLUSIONS

In conclusion, our study identified key sagittal plane kinetic and kinematic differences between the ACLR surgical and non-surgical limbs, and compared to the matched control limb, during a 45° sidestep cut following ACLR and RTS. It appears that post-ACLR participants employed a stiffening sagittal plane movement strategy in the ACLR limb during the stance phase of a 45° sidestep cut, which may increase injury risk and help explain why re-injury rates among post-ACLR individuals is high. These findings did not support our hypothesis that differences would be identified in frontal plane cutting

mechanics between the ACLR surgical limb and the non-surgical and matched control limbs. Control participants did not demonstrate between-limb differences in kinematics with one exception, largely supporting our second hypothesis, and suggesting that our between-limb findings were unique to the ACLR group. This information is important for clinicians to consider when designing and implementing rehabilitation programs and RTS testing for individuals post-ACLR.

## ACKNOWLEDGEMENTS

We would like to acknowledge Justin Ter Har, Brian Newcomb, Jacqueline Diulio, Sherri Dean, Bella Krevitz, Pete Haglund, Morgan Watts, and Hunter Hartman for their assistance with data collection.

### Funding

This work was funded by the Layman Fellowship Program at Oregon State University-Cascades. The funders had no role in study design, data collection and analysis, decision to publish, or preparation of the manuscript.

### Grant Disclosures

The following grant information was disclosed by the authors:
Layman Fellowship Program at Oregon State University-Cascades.

### Competing Interests

The authors declare that they have no competing interests.

### Author Contributions

- Montana Kaiyala performed the experiments, analyzed the data, prepared figures and/or tables, authored or reviewed drafts of the article, and approved the final draft.
- J.J. Hannigan performed the experiments, analyzed the data, prepared figures and/or tables, authored or reviewed drafts of the article, and approved the final draft.
- Andrew Traut performed the experiments, analyzed the data, authored or reviewed drafts of the article, and approved the final draft.
- Christine Pollard conceived and designed the experiments, performed the experiments, analyzed the data, authored or reviewed drafts of the article, and approved the final draft.

### Human Ethics

The following information was supplied relating to ethical approvals (*i.e.*, approving body and any reference numbers):

This study was approved by the Institutional Review Board at Oregon State University.

### Data Availability

The data is available at the Open Science Framework repository: Hannigan, JJ. 2023. "ACL Cutting Project." OSF. December 7. DOI 10.17605/OSF.IO/SZKW2.

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
