# Peer review of "Bilateral movement asymmetries exist in recreational athletes during a 45° sidestep cut post-anterior cruciate ligament reconstruction"

_PeerJ, doi:10.7717/peerj.16948_

## Round 0.1 · original submission · Major Revisions

Both reviewers agree that the manuscript adds new and important knowledge to the field of ACL research. However, they also raised some important points that need to be addressed in the revision of the manuscript.

Reviewer 1 ·

Basic reporting

General Comment:

In the present study, ACLR patients and healthy control subjects were compared regarding biomechanical variables of the knee and hip joint during a 45° cutting maneuver. Lateral comparisons were made in the ACLR patients and healthy controls as well as between groups comparisons using the injured vs. the matched leg.
While I think, the rationales described 45° cutting maneuver as a game-like test analysed using inter and intra individual (leg) comparison are valid and the tables and figures provided are clear and valuable insight for readers, there are major limitations that need to be addressed before considering publication.
Special attention should be given to the rationales of the article and the method. Some important factors, such as approach speed or the comparison between healthy legs (including statistics) are relevant to the content and should, therefore, be reported.

Experimental design

Specific Comments:

Introduction
• The purpose for comparing ACLR patients to healthy controls is not clearly presented. The relevance and news value of this comparison should be elaborated upon.
• Some studies found lateral differences in the hip joint. Why was the hypothesis restricted to the knee joint?
• Explain why the comparison between ACLR patients and healthy control subjects is relevant and the comparison between the legs within healthy subjects not .

Method
Important methodological details are missing:
• How was the ground contact and toe off of the "Cutting maneuver" identified? Ground reaction forces are used for identification in the literature. Please describe this in the method section.
• The "approach speed" measurement method and its non-disclosure need clarification. The exact speed right before the pivotal step is crucial for interpreting your results. Add this data to the article for better interpretation of the data. Ideal for this would be the speed just before the decisive step and not the time for overcoming the distance. (https://doi.org/10.1016/j.jbiomech.2012.06.029.)
• Criteria for matching the legs of healthy subjects and ACLR patients should be clearly stated and reported.

Validity of the findings

Results:
The presentation and interpretation of your results need revising. For instance:
• Display the percentage difference between the legs in the relevant variables to better illustrate physiological relevance.
• Without a comparison of healthy legs between groups, it is difficult to assess whether your results are physiologically relevant to the ACLR patient leg. What would be the relevance of the result if the healthy legs between the groups also showed significant differences ?

Discussion & Conclusion:

• The relevance of a joint angle and excursion difference should be discussed. A percentage representation might be more illustrative.
• The discussion should also consider the significance of comparing the healthy legs of both groups.
• The conclusion doesn't align well with the study's objective.

Additional comments

Additional Point-Specific Comments:

Line 43: Why were only the hip and knee examined?
Lines 52-54: Please clarify the term "avoidance".
Line 73: Why the sudden mention of "non-contact ACL injuries"?
Lines 79-83: This sentence should be shortened or divided for better readability.
Lines 96-97: Please specify what "different demands" means.
Lines 229-233: Elaborate on why the knee experiences increased strain.

Reviewer 2 ·

Basic reporting

Thank you for giving the opportunity to review your interesting manuscript addressing movement asymetries in athletes with and without ACL reconstruction. The introduction is well structured and provides a sufficient overview about ACL specific implications. The methods section provides sufficient information to replicate the study. The Results section briefly described the obtained findings, which are considered in the Discussion Section. As I missed mentioning unanticipated conditions in the introduction, I found this aspect in the limitations sections, where it fits well. Nevertheless, I will provide some suggestions and comments for potential consideration.
Lines 88 – 95: In this section you alltime used the wording authors found that… This comment is applicable to further sections of your manuscript, e.g. lines 265 – 271 in the Discussion where the phrasing was used about 5 times in a row. Please revise.

Lines 101 – 103: I am confused by your first hypothesis. While I do understand the hypothesis regarding the knee valgus angle (reduced hip peak extensor strength) I do not get the peak knee varus moment. A higher peak varus moment would counteract the knee valgus, doesn’t it? Please consider a rephrasing to clarify and avoid confusion.

Experimental design

Lines 119 – 120: Was the sample size calculation performed via G-Power? Which data collection are you referring to? Please consider including a reference.

Lines 176 – 185: You mentioned using the unpaired t-test as an ANOVA would violate the assumption of independency of the observations. However, in fact, the uninjuried leg and injuried leg in the group of ACLR patients are not independent, as they are part of one and the same body. Using the 2-way ANOVA might mix up the effects of independent (including the uninjuried and injuried leg of the ACLR group and both legs of the control). However, in contrast, what do you think about alpha error accumulation by using several t-tests? Additionally you would have to perform t-tests for leg a and b, a and c, and d, b and c, b and d, c and d. As well as the paired t tests and unpaired t test for pre-test conditions for both groups.

Validity of the findings

“It was hypothesized that a) post-ACLR individuals would cut with a greater peak knee valgus angle and peak knee varus moment in their involved limb compared to their uninvolved limb and the matched control limb, and b) control participants would not demonstrate any frontal or sagittal plane between-limb differences. “
Lines 213 – 224: The listed aspects in the first part of the discussion were not part of your hypotheses. Therefore, I wonder about mentioning knee flexion, knee extension, stiffed landing kinematics etc., as you just hypohtsized differences about knee valgus/varus and differences in the control groups legs. Please clarify

Lines 244,245: You described decreased quadriceps strength to possibly be responsible for your listed findings (reduced peak knee flexion and knee extensor moment). Firstly, reduced quadriceps strength might not reduced peak knee flexion moment ? Secondly, in line 246, you describe that higher forces of the quadriceps would be applied to the ACL in a more extended position. I am confused; what is correct? Reduced quadriceps strength or higher forces from the quadriceps applied to the ACL.
I can imagine the intention of the sentence however, to me it should be described more concisely.
Additionally, apart from more favorable muscle lengths for force production, you should not forget the limited force production capacity in stretched muscle lengths of the hamstring muscles with a more extended knee joint (stiffed landing). Therefore, a imbalance between quadriceps strength and hamstring strength could be hypothesized as well.
In line 256 you use an abbreviation for vertical ground reaction forces. If I did not miss anything, this word does not appear again in your manuscript. Please only use abbreviations if you use the word afterwards frequently in the text.
Line 271 – 277: this sentence is a little long. Please consider splitting it into at least 2.
Limitations:
Line 278 – 282: It was not the formulated hypothesis to test knee kinematics under unanticipated conditions. Therefore, this was no limitation of the study. Otherwise you could include thousands of “limitations”, as you did not text all parameters you could. However, what were the limitations of the study when addressing your initially formulated hypotheses? You could list this as well in the last sentence, which seems to be a kind of an outlook, requesting the inclusion of underlying mechanisms in future research as well as unplanned conditions.
Conclusions
Please summarize the results with a focused answer regarding you research hypotheses. You did not formulated that there would be differences between the limbs as well as the ACLR group and the others. You hypothesized differences in the knee valgus/varus, however, I do not find a conclusion answering the corresponding research questions.

Additional comments

I hope some of the suggestions will help you improving parts of your manuscript, thanks again for providing your work and effort, best wishes.

---

## Round 0.2 · Minor Revisions

Please note the final comments from Reviewer 1 before the manuscript will be accepted.

Reviewer 1 ·

Basic reporting

no comment

Experimental design

no comment

Validity of the findings

The article lacks clarity in reporting the specific tests that yielded significant results. It is essential to clearly indicate whether the significance was determined through the Linear Mixed-Effects Model (LMM) or the subsequent Bonferroni-corrected t-tests. To enhance transparency including F/T and df values for both the model and t-tests in the Results section and tables is recommended. This will provide readers with a comprehensive understanding of the statistical significance of your findings.
Please modify the result section and associated tables.

Additional comments

52: The p-value for uniformity should be added to the Abstract.

75-81: What does this mean for the rehabilitation process? You want to express that the rehab is obviously not very successful or that the injury cannot be completely rehabilitated. Please write a concluding sentence

99: The abbreviation COD has already been introduced

If I am not mistaken, there is a uniformity in biomechanics that internal joint moments are reported. Therefore, the mention of internal / external in the introduction / discussion is not necessary. I.e. it should only indicate deviations from this concept.

117: It could be highlighted that these are "matched" control participants, which is a strength of the study.

172-175: If participants were not explicitly instructed to run at 3.35 m/s, you should clarify this information and describe the actual procedure accurately.

184-185: You may omit this passage and instead present the Approach Speed (Mean ± Standard Deviation) in the Results.

Reviewer 2 ·

Basic reporting

All comments were addressed. Even though I would recommend a more detailed reporting of the adjusted statistics, this might be a matter of taste. Therefore, I think it is up to the authors to further improve their manuscript, if they feel the necessity of addressing this issue.

Furthermore, I feel the discussion could be more concise and better structured. Since those are, from my point of view, subjective impressions of mine, I will not delay publishing their results.

Experimental design

no comment

Validity of the findings

no comment

Additional comments

no comment

---

## Round 0.3 · accepted · Accept

All reviewer comments have been addressed and the manuscript is now ready for publication.